# Comparative Study of Oil Recovery Using Amphoteric Terpolymer and Hydrolyzed Polyacrylamide

**DOI:** 10.3390/polym14153095

**Published:** 2022-07-29

**Authors:** Iskander Sh. Gussenov, Nurbatyr Mukhametgazy, Alexey V. Shakhvorostov, Sarkyt E. Kudaibergenov

**Affiliations:** 1Institute of Polymer Materials and Technology, Almaty 050019, Kazakhstan; nurbatyr.kaz@gmail.com (N.M.); alex.hv91@gmail.com (A.V.S.); 2Department of Petroleum Engineering, Satbayev University, Almaty 050013, Kazakhstan

**Keywords:** ternary polyampholyte (TPA), hydrolyzed poly (acrylamide) (HPAM), antipolyelectrolyte effect, sand-pack model, water flooding, polymer flooding, oil recovery factor (ORF)

## Abstract

This paper presents the viscosifying and oil recovery efficiencies of a novel high-molecular-weight ternary polyampholyte (TPA), composed of 80 mol.% acrylamide (AAm) (a nonionic monomer), 10 mol.% 2-acrylamido-2-methyl-1-propanesulfonic acid sodium salt (AMPS) (an anionic monomer), and 10 mol.% (3-acrylamidopropyl) trimethylammonium chloride (APTAC) (a cationic monomer), in various high-salinity brines as compared to the efficiency of hydrolyzed poly(acrylamide) (HPAM), which is the most commonly used polymer in oil production. The results show that, in a range of salinity from 200 to 300 g∙L^−1^, the viscosity of the TPA solution is rather high and relatively stable, whereas that of HPAM severely decreases. The ability of TPA to increase its viscosity in extremely high salinity brines is explained by the antipolyelectrolyte effect, resulting in the unfolding of macromolecular chains of charge-balanced polyampholytes at a quasi-neutral state, which occurs due to the screening of the electrostatic attraction between oppositely charged moieties. The novelty of this research is that, in high-salinity reservoirs, the amphoteric terpolymer Aam-AMPS-APTAC may surpass HPAM in oil displacement capability.

## 1. Introduction

Water flooding is the most widely used oil recovery method. However, in many reservoirs, water cannot provide a stable oil displacement front due to the unfavorable mobility ratio between these fluids, which results in viscous fingering, especially in high-viscosity oil reservoirs. Moreover, permeability heterogeneities can also cause premature water breakthrough, and, as a result, less oil is produced than would have been possible with better conformance control [1]. Such problems can be solved by increasing the water viscosity, ergo water-soluble polymers are used for this purpose, usually at low concentrations of 0.5–2.5 g∙L^−1^, as they provide viscosity values that are high enough to effectively push the oil out via polymer flooding [2].

Hydrolyzed polyacrylamide (HPAM) is the most widely applied polymer for the following reasons:Its low cost;Its commercial availability;Its remarkable capability to increase viscosity;Its acceptable and adjustable injectivity;Its resistance to microbial degradation.

However, the viscosity of HPAM decreases as brine salinity increases [3]. Figure 1 shows the dependence of the viscosity of Pusher 1000, a commercial HPAM, versus brine salinity [4]. As can be seen, the viscosity of HPAM decreases from 100 cp to 4.5 cp upon a rise in NaCl content from 0.05 to 8 wt.%. Here, 8 wt.% NaCl corresponds to 84 g∙L^−1^ salinity, which is less than the salinity of many oilfield brines, such as those that are found in Kazakhstan. In most polymer flooding projects, the viscosity of the polymer solution is higher than 20 cp [2]; thus, high HPAM concentrations may be required to allow for acceptable EOR. For example, in low-salinity conditions, 50% less polymer is needed to achieve the target mobility ratio [5]. A highly concentrated polymer solution makes polymer flooding projects economically unattractive.

One of the fundamental properties discovered in the last century with respect to linear and crosslinked polyampholytes is the antipolyelectrolyte effect [6,7,8,9,10]. The basis of the antipolyelectrolyte effect is the unfolding, or swelling, of macromolecular chains of annealed (pH-dependent) polyampholytes at the isoelectric point (IEP) and of charge-balanced (pH-independent) polyampholytes in a quasi-neutral state in salt solution, which occurs due to the screening of the electrostatic attraction between oppositely charged moieties. The unfolding of charge-balanced polyampholytes in a quasi-neutral state with an increasing concentration of salt is in full accordance with the theory developed by Khokhlov et al. [11]. The difference between the polyelectrolyte and antipolyelectrolyte effects upon the addition of salt is illustrated in Figure 2 [12].

The solution properties of quenched polyampholytes based on a fully charged anionic monomer, the sodium salt of 2-acrylamido-2-methyl-1-propanesulfonic acid (AMPS), and a cationic monomer, (3-acrylamidopropyl) trimethylammonium chloride (APTAC), have been studied in aqueous salt solutions [13]. The behavior of quenched diblock polyampholytes, composed of an anionic block (AMPS)_82_, in which the lower subscript “82” indicates the degree of polymerization, and cationic blocks (APTAC)_n_, in which *n* = 37, 83, and 181 (abbreviated as AMPS_82_-*b*-APTAC_n_), has been evaluated in aqueous and aqueous salt solutions [14]. Equimolar AMPS_82_-*b*-APTAC_83_ has been shown to precipitate in pure water, due to the formation of a polyion complex between anionic and cationic blocks, and to dissolve upon the addition of NaCl, demonstrating the antipolyelectrolyte effect. In contrast, AMPS_82_-*b*-APTAC_37_ and AMPS_82_-*b*-APTAC_181_, containing an excess of anionic and cationic blocks, are fully water-soluble. Thus, these polyelectrolyte chains unfold in pure water but coil in salt solution due to the screened electrostatic repulsion between uniformly charged macroions. Such examples clearly show that the selected ternary polyampholytes, based on Aam-AMPS-APTAC, are expected to exhibit the antipolyelectrolyte effect and be effective viscosifiers in saline solution so as to enhance oil recovery.

The ability of low-charge-density amphoteric copolymers and terpolymers to swell and be effective viscosity enhancers in high-salinity and high-temperature reservoirs plays a crucial role in EOR [15]. Polyampholytes, due to their resistance to high salinity and temperatures, are under consideration as viscosifying agents in situations where thickeners are required in concentrated brine solutions [16,17,18,19,20,21,22]. Polyampholytes with a hydrophobic functional group may be especially effective viscosity enhancers in high-salinity media and at high temperatures, as they have both a self-assembly ability and an adjustable charge balance [23,24,25,26]. The application of acrylamide-based polyampholytes in EOR and drag reduction was reviewed by Rabiee et al. [27]. In this regard, amphoteric polyelectrolytes, which have both cationic and anionic groups, are an interesting object for research, because the attraction between positively and negatively charged groups is screened out by low-molecular-weight ions in brines, causing the polymer chain to expand, leading to a viscosifying effect [12,28,29].

Previously, the current project team synthesized a series of TPAs, [AAm]:[AMPS]:[APTAC] (50:25:25; 60:20:20; 70:15:15; 80:10:10, and 90:5:5 mol.%), and tested their suitability for EOR [29]. Only the 80:10:10 mol.% sample was found to increase oil recovery, by up to 4.8–5%, under oil reservoir conditions (with a salinity of 163 g∙L^−1^ and a temperature of 60 °C), the low value of the oil recovery of which was attributed to the low molecular weight of the samples of [AAm]:[AMPS]:[APTAC] in general. In a further study [30], a high-molecular-weight terpolymer, [AAm]:[AMPS]:[APTAC] (80:10:10 mol.%), with a weight-average molecular weight of M_w_ = 2.9 × 10^6^ Dalton and an average-number molecular weight of M_n_ = 2.1 × 10^6^ Dalton, was synthesized and characterized using various physicochemical methods. The dynamic and reduced viscosities of this ratio of the TPA were found to increase in saline water due to the antipolyelectrolyte effect, the mechanism of which is comprehensively described above. An injection of 0.25 wt.% TPA solution prepared in 200 g∙L^−1^ brine resulted in increases in incremental oil recovery of 28 and 23% in 0.62 and 1.8D sand-pack models, respectively. However, this project did not provide any comparison between TPA and HPAM, because the latter is already the most studied and used polymer in EOR. The present article compares the efficiency of the two polymers regarding their viscosity and sand-pack flooding. As is demonstrated in the Results and Discussion Section of this paper, the main difference between the HPAM and TPA is that the former is a polyelectrolyte, with viscosity that drastically decreases in salt solution (see Figure 1), whereas the latter is a polyampholyte, with viscosity that increases in saline water due to the antipolyelectrolyte effect. Seemingly, the advantage of the antipolyelectrolyte effect in the case of the ternary polyampholyte over the polyelectrolyte effect of HPAM has not yet been evaluated in sand-pack flooding tests. The novelty of the current research is that the ternary polyampholyte based on AAm-AMPS-APTAC may have a superior oil displacement capability compared to that of HPAM in high-salinity reservoirs. Thus, the results of these preliminary experiments may encourage petroleum engineers to scale up such polyampholytes for application in field conditions.

Other relatively novel methods are available to improve the properties of viscous oils and make them more mobile. For example, the method described by Zhou et al. [31] allows for the reduction of the viscosity of oil by up to 87.32%. Polymer flooding is still a widely applied method in the recovery of heavy oils [32]. Moreover, the synthesis and study of new polymers is a relevant task, as, in addition to EOR, they are used in other important applications, such as improving wellbore stability [33].

## 2. Experimental Part

### 2.1. Materials

#### 2.1.1. Polymers

TPA (AAm-AMPS-APTAC), the synthesis and characterization of which has already been comprehensively described by the current research team [30], contains repeating monomer units consisting of 80 mol.% AAm, 10 mol.% AMPS, and 10 mol/% APTAC (Figure 3).

Firstly, the dry acrylamide (AAm) monomer was dissolved in distilled water while being stirred in a beaker. Then, the liquid monomers, AMPS and APTAC, were added in equimolar concentrations to the aqueous solution of AAm. The total concentration of the monomers in the water was equal to 50 wt.%. The monomer mixtures were purged with nitrogen over 15 min to remove the dissolved oxygen. Afterward, the red–ox pair catalyst, composed of ammonium persulfate and sodium bisulfite, was added to initiate radical polymerization. The polymerization process, accompanied by increasing viscosity, occurred over 4 h. The as-prepared ternary polyampholyte was used in oil displacement experiments without further purification.

The weight-average molecular weight (M_w_) and the average-number molecular weight (M_n_) of the terpolymer, as determined by gel-permeable chromatography, were equal to 2.9 × 10^6^ and 2.1 × 10^6^ Dalton, respectively. Hydrolyzed polyacrylamide (HPAM), with a hydrolyzation degree of 30% and an average molecular weight of 17.2 × 10^6^ Dalton (Flopaam 3630 S, 98% purity, SNF), was used for a comparative study.

#### 2.1.2. Sand-Pack Models

Two types of sand packs were used, the diameter and length of which were equal to 3 by 5 cm and 4.3 by 8 cm, respectively. The grain size of the quartz sand used in this study ranged between 0.25 and 0.5 mm.

#### 2.1.3. Core Samples

Aerated concrete was used to prepare the core samples (Figure 4), which are characterized by extremely high porosity and permeability.

#### 2.1.4. Brine

The salinity and chemical composition of the different brines that were used in the present work are listed in Table 1. Furthermore, East Moldabek reservoir brines, with salinity equal to 100 and 163 g∙L^−1^, were used for sand-pack and core flooding tests.

#### 2.1.5. Oil

The following different types of crude oils were used in this study, with the temperatures approximating those recorded at the oil wells from which the samples were taken:

East Moldabek (oil well 2027) with a density and viscosity equal to 0.89 g/cm^3^ and 138.8 cp, respectively, at 25 °C.

Karazhanbas (oil well 1913) with density values of 0.93 g/cm^3^ and 0.907 g/cm^3^ at 30 °C and 60 °C, respectively, while the viscosity corresponds to 420 cp and 64 cp at 30 °C and 60 °C, respectively.

### 2.2. Methods

#### 2.2.1. Rheological Studies

The viscosities of the TPA and HPAM were measured using a Brookfield viscometer (Spindle-0) at a stable shear rate of 7.32 s^−1^. The temperature in the rheological tests varied from room temperature (24 °C) to 60 °C. However, an Anton Paar Rheolab QC viscometer was used to measure the viscosity of the effluent samples for the HPAM experiment on the 8.6 cm-length sand pack.

#### 2.2.2. Sand Pack and Core Flooding

Sand pack flooding was conducted with the help of a special core flooding setup equipped with a Hassler core holder and piston pumps (Figure 5).

The general methodology of the sand-pack and core flooding tests is described in the following steps:Vacuum the model for at least 10 min;Saturate the model with brine using a high-pressure piston pump;Displace brine with oil until irreducible water saturation is reached;Simulate water flooding using one or, in some cases, more pore volumes of brine;Simulate polymer flooding using several pore volumes of TPA or HPAM solutions.

The flow rates, ambient temperature, and confining pressure values used in the tests are outlined in Section 3.2. Of importance is that, in this study, the injection pressure was measured only at the inlet across the injection tube, whereas the outlet of the core was open to the atmosphere.

The effluent samples containing oil and brine/polymer were collected in glass test tubes. The fluids were separated manually by withdrawing water from the test tube using a syringe. The mass of oil displaced from the model was measured by calculating the difference between the masses of the test tubes with and without oil. Finally, the oil recovery factor was calculated by dividing the mass of the displaced oil by the initial mass of the oil in the model.

## 3. Results and Discussion

### 3.1. Rheological Studies

Figure 6 shows that the viscosity of the 0.25 wt.% TPA solution is relatively stable and equal to around 26 cp at 24 °C and around 14 ± 1 cp at 60 °C in brine with salinity from 200 to 300 g∙L^−1^. In contrast, the viscosity of the 0.25 wt.% HPAM solution decreases from 28.25 cp to 7.5 cp at 24 °C and from 11.2 cp to 3.25 cp at 60 °C in brine with salinity from 200 to 300 g∙L^−1^.

The structures of HPAM, a polyelectrolyte, and TPA, a polyampholyte, dictate their behaviors in saline solutions. The HPAM chains tend to coil with an increase in salinity, because the electrostatic repulsion between negatively charged carboxylic groups is screened by the added salts, demonstrating the polyelectrolyte effect. Moreover, the bivalent cations (Ca^2+^ and Mg^2+^) present in saline solution can bridge the carboxylic ions in the HPAM, effectively shrinking the macromolecules [34]. However, the TPA macromolecule chains tend to unfold in saline solutions due to the screening of the electrostatic attraction between negatively charged AMPS and positively charged APTAC moieties, demonstrating the antipolyelectrolyte effect.

The rheological results show that the viscosity of the 0.25 wt.% HPAM solution in 200 g∙L^−1^ decreases over 15 days of aging by 27.5%, whereas that of the 0.25 wt.% TPA solution in 200 g∙L^−1^ decreases over the same time period by only 18.2% (Table 2 and Figure 7).

### 3.2. Core and Sand-Pack Flooding Tests

#### 3.2.1. Experiment 1

The flooding of the 8.6 cm-long high permeability (15.8 Darcy) sand pack saturated with East Moldabek oil and 100 g∙L^−1^ reservoir brine at room temperature demonstrated that the 0.2 wt.% HPAM solution in low-salinity brine (15 g∙L^−1^ NaCl), with initial viscosity equal to 31 cp at 14.7 s^−1^, provides a notable increase in oil recovery (5–6%), even after the injection of 3 PVs (1 PV = 64 cm^3^) of pre-flush with 1.7 PV of 100 g∙L^−1^ brine and 1.3 PV of various TPA solutions, which themselves were not effective in oil displacement (Figure 8).

This is a notable result, because the injection of 3 PVs of fluid into the homogeneous sand pack drives the model almost to its irreducible oil saturation value, at which point the incremental oil recovery increase of 5–6% proves that HPAM has potential application as an effective polymer for EOR when dissolved in low-salinity brine. Moreover, an analysis of the effluent samples shows that, after the injection of 1 and 2 PVs of the HPAM solution, the viscosities of the effluents rose to 28.5 cp and 29.8 cp, respectively (Figure 9). These correspond to 8% and 3.8% viscosity reductions, respectively, in comparison with the initial value of 31 cp, demonstrating the good propagation ability of HPAM in a high-permeability sand pack.

#### 3.2.2. Experiments 2 and 3

The next two experiments were conducted using high-porosity (83–85%) aerated concrete cores (4.4–4.5 cm long and 2.9 cm diameter) with permeabilities of 5.06 Darcy and 4.72 Darcy in the second and third experiments, respectively. The 0.5 wt.% TPA and HPAM solutions were used in the second and third experiments, respectively, with both dissolved in 163 g∙L^−1^ reservoir brine and injected at 60 °C and 1 cm^3^/min into cores previously saturated with Karazhanbas oil and the aforementioned brine. Preliminarily, the cores were subjected to an injection of 1 PV of water. Figure 10 shows the mass of the oil produced versus the total mass of all the other fluids for both the TPA and HPAM experiments. As can be seen, the water flooding results are very similar for both experiments. However, HPAM, at its maximum, allowed the displacement of 3 times more oil than did TPA. The photos of the core inlet (face) and outlet (back) show that the core used in the HPAM experiment contains less oil than that from the TPA test, especially at the outlet side. Moreover, Figure 11 shows that the pressure drop of HPAM was around 3 times higher than that of TPA. This demonstrates that the apparent viscosity of HPAM is higher, and it explains its better performance in terms of greater oil production.

#### 3.2.3. Experiments 4 and 5

These tests were conducted using 0.25 wt.% polymer solutions prepared in 200 g∙L^−1^ brine injected into sand packs (5 cm in length and 3 cm diameter) saturated with Karazhanbas oil and 200 g∙L^−1^ brine at 30 °C and 0.15 cm^3^/min. Table 3 shows the stabilized oil pressure measured at irreducible water saturation during the process of oil injection. Prior to the polymer injection, the models were subjected to 1 PV of 200 g∙L^−1^ brine injection in order to simulate the water flooding process (Table 3). The pressure data presented in Table 3 were used to calculate the mobility ratio in order to assess the initial conditions for polymer flooding. The mobility ratio was lower in the case of HPAM, which makes the conditions more favorable for polymer flooding. However, as can be seen in Figure 12, TPA, at its maximum, allowed 2 times more displacement of oil than did HPAM, even though the conditions in the HPAM experiment were more favorable for polymer flooding.

Moreover, the apparent viscosity of the TPA solution was higher than that of the HPAM solution, which is demonstrated by the notably higher pressure drops of around 2 times registered during the injection of the TPA solution (Figure 13), thus explaining why TPA produced more oil than did HPAM.

As can be seen in Figure 14, the total mass of oil produced in the course of TPA and HPAM injections is around of 27% and 18%, respectively, in comparison with the initial oil mass in the sand packs. The injection of 13–35 cp 15.2 × 10^6^ g/mol HPAM solution into 200–600 mD cores (10 cm long), saturated with 10.2 cp oil and flooded with low-salinity brine, allowed for the displacement of 15–25% of the oil [35]. In contrast, the injection of 2 PVs of 55 cp HPAM solution permitted the displacement of 26.2% of the oil after water flooding. The injection of 3 PVs of 6–52 cp 21 × 10^6^ g/mol HPAM solution into the 206–410 mD cores (6.5 cm long), saturated with 990–1610 cp oil and flooded with 1.5 PV of 1.9% TDS brine, allowed for the displacement of around 30% of the oil after water flooding [32]. Thus, the results shown in Figure 14 are within the range found in the literature for a wide variety of oils.

## 4. Conclusions

A novel high-molecular-weight amphoteric terpolymer (TPA), composed of 80 mol.% acrylamide (AAm) (a nonionic monomer), 10 mol.% 2-acrylamido-2-methyl-1-propanesulfonic acid sodium salt (AMPS) (an anionic monomer), and 10 mol.% (3-acrylamidopropyl) trimethylammonium chloride (APTAC) (a cationic monomer), was successfully tested as an oil displacement agent for polymer flooding experiments. The rheological studies demonstrated that the viscosity of TPA is stable in the range of 200–300 g∙L^−1^ brine salinity at 24 and 60 °C, whereas the viscosity of the HPAM solution drops significantly upon increasing the salinity. Moreover, the TPA solution was shown to be more resilient to viscosity reduction after aging at room temperature than was HPAM. Indeed, following 15 days of aging, a viscosity reduction of 18.2% occurred in TPA, which is significantly lower than the 27.5% in HPAM. Sand-pack and core flooding experiments provided valuable results. In particular, the efficiency of HPAM at relatively low salinity (15 g∙L^−1^ NaCl) was proven by its better incremental oil recovery of 5–6% after pre-flushing the model with 3 PVs of brine and various TPA solutions, which themselves had proven comparatively ineffective at that level of salinity. In the core flooding tests, HPAM also outperformed TPA in 163 g∙L^−1^ brine, which was considered to be moderate salinity for this study. However, when the brine salinity was increased to 200 g∙L^−1^, the TPA allowed the production of 2 times more oil at its maximum than did HPAM. This finding adds to the novelty and significance of this work and is explained by the higher apparent viscosity of the TPA solution in the porous media, which was indicated by pressure drops of 2 times. Therefore, ternary polyampholyte (TPA), having both cationic and anionic groups, resulting in the attraction between positively and negatively charged groups being screened out by low-molecular-weight ions in brines, causes the polymer chain to expand, leading to a viscosifying effect in clear demonstration of the benefits of the antipolyelectrolyte effect on EOR.

## Figures and Tables

**Figure 1 polymers-14-03095-f001:**
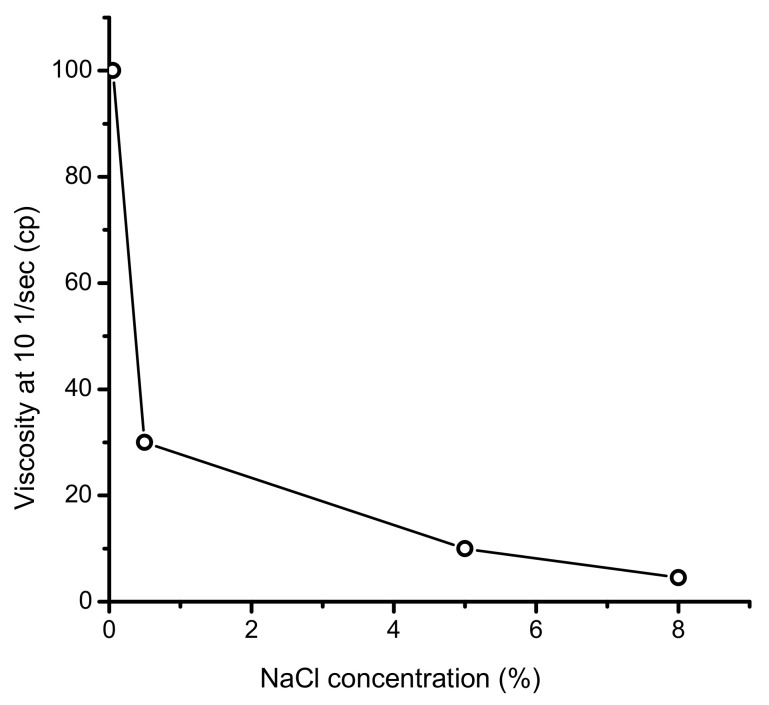
Viscosity of 1.1 × 10^−7^ mol/kg of Pusher 1000, a commercial HPAM solution, at 28.5 °C, adapted from Samanta et al. [4].

**Figure 2 polymers-14-03095-f002:**
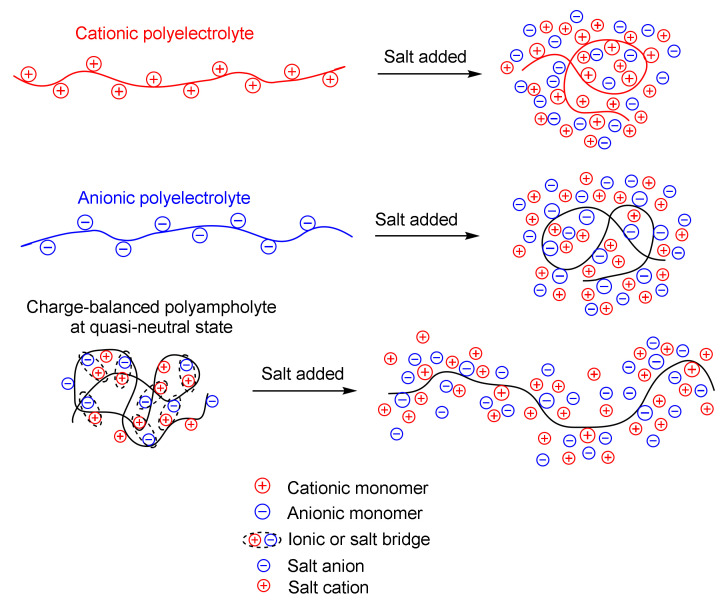
Illustration of the behavior of polyelectrolytes and polyampholytes in a salt solution. Reprinted with permission from Ref. [12]. 2021, Sarkyt E. Kudaibergenov [12].

**Figure 3 polymers-14-03095-f003:**
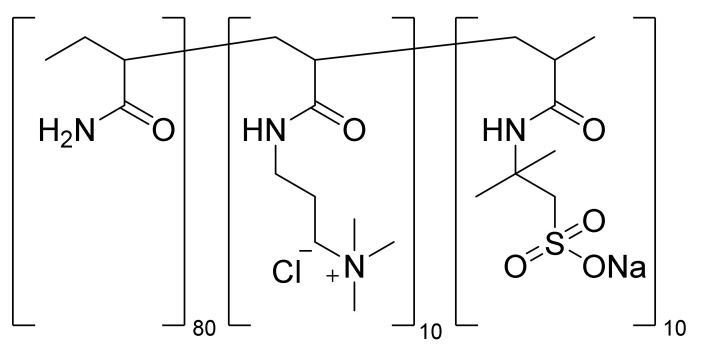
Repeating monomeric units and composition of TPA (AAm-AMPS-APTAC).

**Figure 4 polymers-14-03095-f004:**
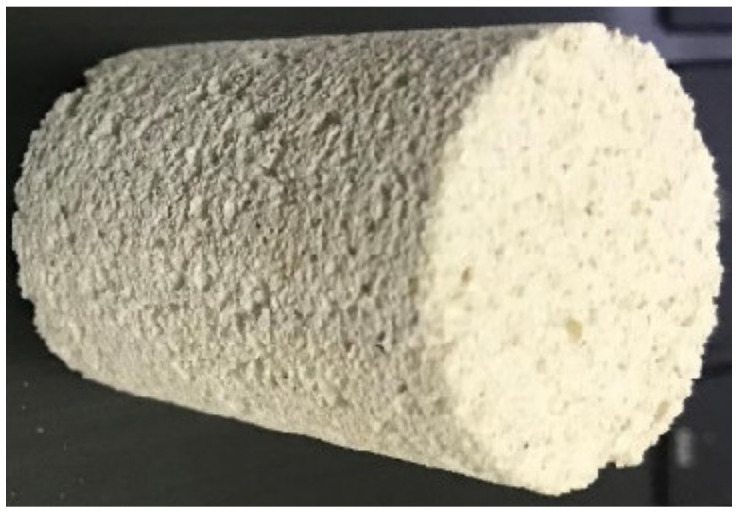
Aerated concrete core sample.

**Figure 5 polymers-14-03095-f005:**
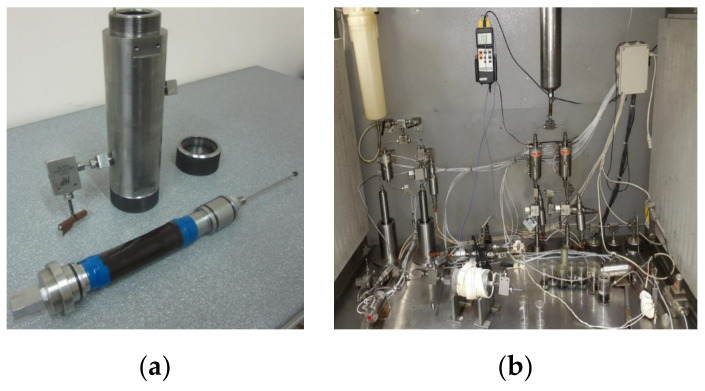
Hassler core holder (**a**) and core flooding setup (**b**) used in the present study.

**Figure 6 polymers-14-03095-f006:**
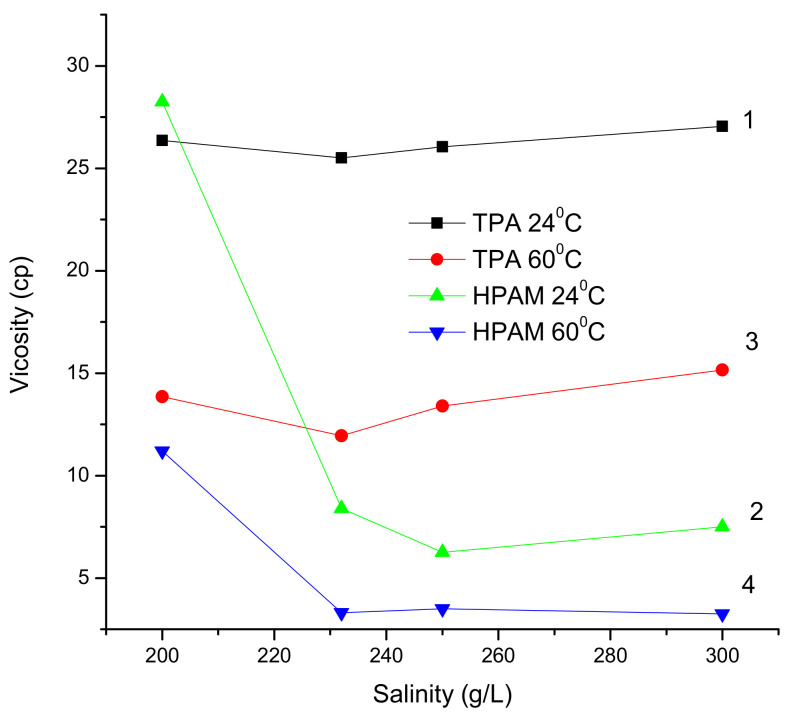
Dynamic viscosities of 0.25 wt.% TPA (curves 1 and 3) and HPAM (curves 2 and 4) in saline solutions at 24 °C (curves 1 and 2) and 60 °C (curves 3 and 4), with a shear rate of 7.32 s^−1^.

**Figure 7 polymers-14-03095-f007:**
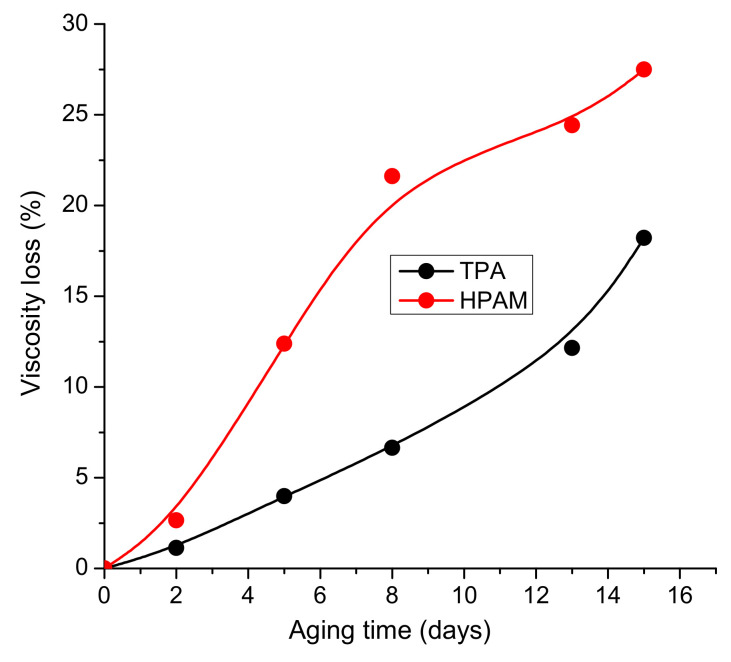
Viscosity loss for 0.25 wt.% TPA and HPAM solutions in 200 g∙L^−1^ synthetic brine while aging at 24 °C.

**Figure 8 polymers-14-03095-f008:**
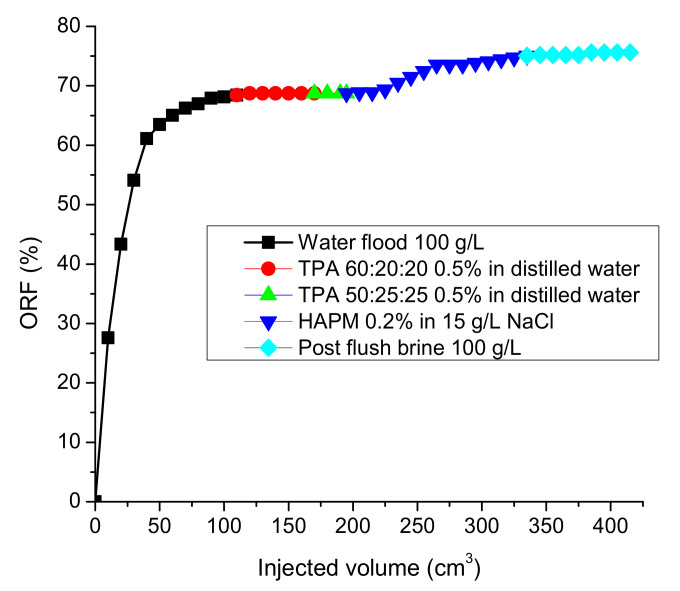
Oil recovery factor versus injected volume, with flow rate set at 0.1 cm^3^/min.

**Figure 9 polymers-14-03095-f009:**
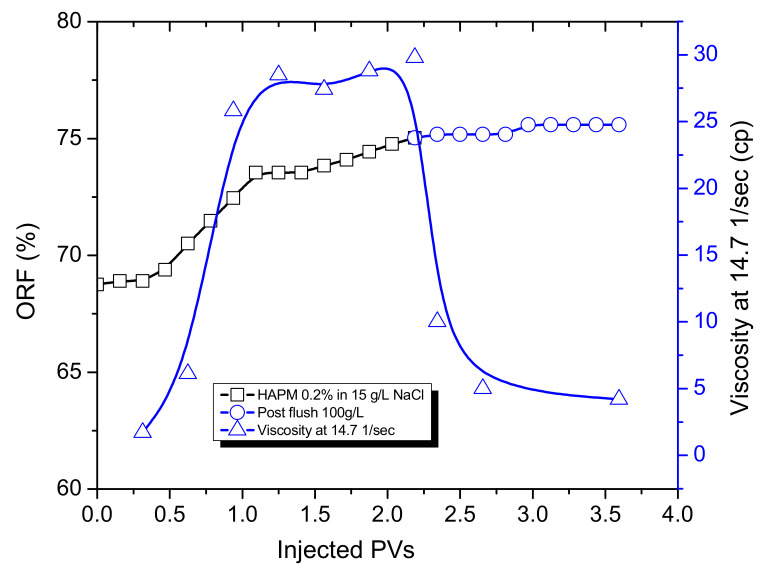
Viscosity of effluents and ORF versus injected PVs of HPAM solution. HPAM 0.2 wt.% with initial viscosity of 31 cp at 14.7 s^−1^.

**Figure 10 polymers-14-03095-f010:**
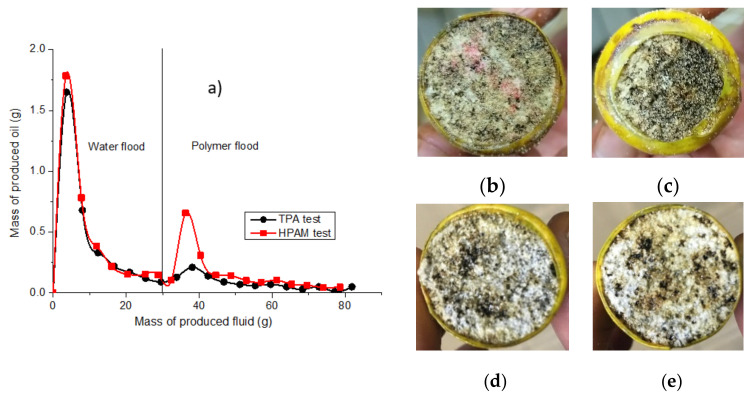
(**a**) Mass of oil produced during water and polymer injections in the aerated concrete cores; core inlet (**b**) and outlet (**c**) after testing with TPA; core inlet (**d**) and outlet (**e**) after testing with HPAM.

**Figure 11 polymers-14-03095-f011:**
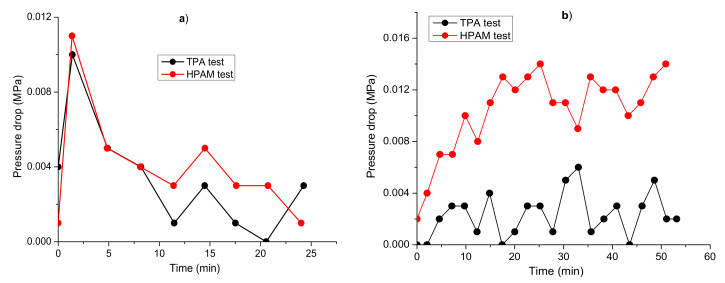
Pressure drops registered during flooding tests in aerated concrete cores with TPA and HPAM for water (**a**) and polymer (**b**) floods.

**Figure 12 polymers-14-03095-f012:**
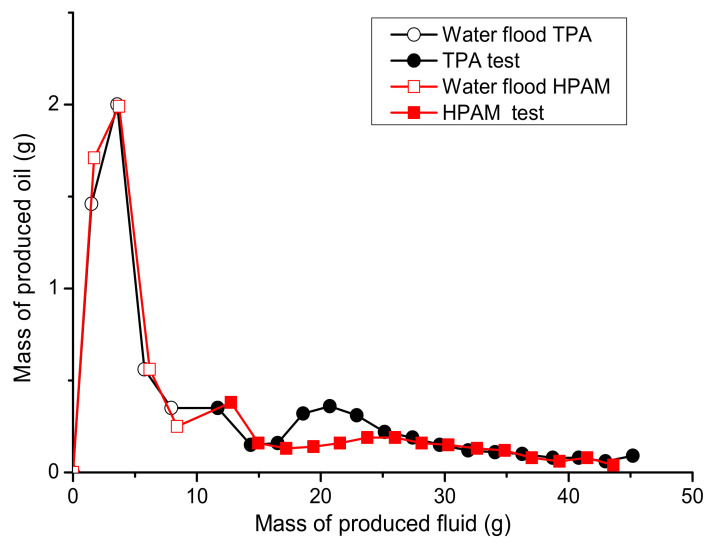
Mass of oil produced during water and polymer injections into the sand-pack models during experiments 3 and 4.

**Figure 13 polymers-14-03095-f013:**
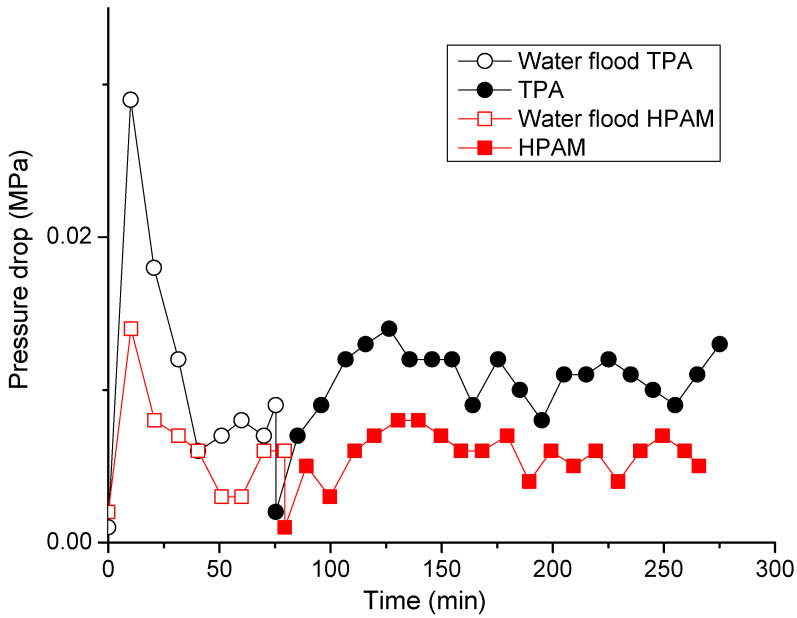
Pressure drop values registered during the water and polymer flooding experiments, using 5 cm-long sand packs and 0.25 wt.% TPA and HPAM solutions in 200 g∙L^−1^ brine.

**Figure 14 polymers-14-03095-f014:**
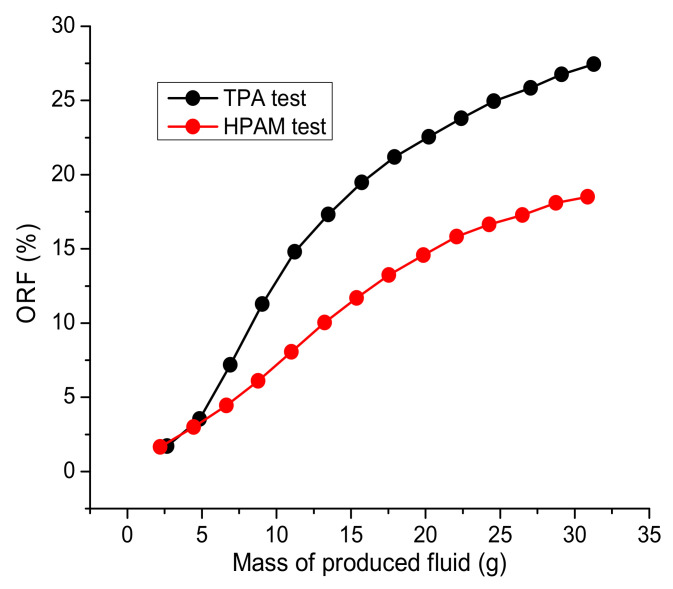
Oil recovery factor versus the mass of fluid produced as registered during the water and polymer flooding experiments, using 5 cm-long sand packs and 0.25 wt.% TPA and HPAM solutions in 200 g∙L^−1^ brine.

**Table 1 polymers-14-03095-t001:** Chemical composition of synthetic brines.

Salinity (g∙L^−1^)	Concentration of Salts (g∙L^−1^)		
	NaCl	CaCl_2_	MgCl_2_
200	180	10	10
232	208.8	11.6	11.6
250	225	12.5	12.5
300	270	15	15

**Table 2 polymers-14-03095-t002:** Viscosity of 0.25 wt.% TPA and HPAM solutions in 200 g∙L^−1^ synthetic brine while aging at 24 °C.

Date	Aging Time, Days	Dynamic Viscosity, (±1 cp)	
		TPA	HPAM
6 October 2021	0	26	28
8 October 2021	2	26	27
11 October 2021	5	25	25
14 October 2021	8	25	22
19 October 2021	13	23	21
21 October 2021	15	22	20

**Table 3 polymers-14-03095-t003:** Pressure data measured during oil saturation and water flooding.

Experiment	Polymer Used	Stabilized Oil Pressure Drops (MPa)	Stabilized Water Pressure Drops (MPa)	Mobility Ratio
4	TPA	0.044	0.005	8.8
5	HPAM	0.02	0.005	4

## Data Availability

Not applicable.

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
