# Peer review of "Comparative Study of Oil Recovery Using Amphoteric Terpolymer and Hydrolyzed Polyacrylamide"

_polymers, 2022, doi:10.3390/polym14153095_

Round 1

Reviewer 1 Report

The present study aims to evaluate the viscosifying and oil recovery efficiency of a novel high molecular weight amphoteric terpolymer (ATP), composed of 80 mol.% of acrylamide (AAm), a nonionic monomer, 10 mol.% of 2-acrylamido-2-methyl-1-propanesulfonic acid sodium salt (AMPS), an anionic monomer, and 10 mol.% of (3-acrylamidopropyl) trimethylammonium chloride (APTAC), a cationic monomer, in various high salinity brines. The results were compared with those of hydrolyzed poly(acrylamide) (HPAM), which is the most widely used polymer in oil production.

The present study is well elaborated with a clear description of the state of the art. The Introduction part provides significant information regarding the scope of the present research.

Some remarks:

1.  In Figure 4 please insert the sign "degree" in the figure legend.

2. In the Results and Discussion part is is better to compare the obtained results with another ones from literature.

Author Response

Dear Editor,

Please find enclosed our responses to reviewer’s comments and revised manuscript. Using this opportunity, we would like to express our sincere thanks to all reviewers for valuable comments and suggestions, which have helped us substantially to improve the previously submitted manuscript.

Reviewer 1

The present study aims to evaluate the viscosifying and oil recovery efficiency of a novel high molecular weight amphoteric terpolymer (TPA), composed of 80 mol.% of acrylamide (AAm), a nonionic monomer, 10 mol.% of 2-acrylamido-2-methyl-1-propanesulfonic acid sodium salt (AMPS), an anionic monomer, and 10 mol.% of (3-acrylamidopropyl) trimethylammonium chloride (APTAC), a cationic monomer, in various high salinity brines. The results were compared with those of hydrolyzed poly(acrylamide) (HPAM), which is the most widely used polymer in oil production.

The present study is well elaborated with a clear description of the state of the art. The Introduction part provides significant information regarding the scope of the present research.

Some remarks:

  1. In Figure 4 please insert the sign "degree" in the figure legend.

Answer:

Figure 4 became figure 5 and has been corrected.

  1. In the Results and Discussion part is better to compare the obtained results with another ones from literature.

Answer:

The comparison with the oil recovery data presented in the literature is provided at the end of the section. Please see page 12, lines 315-322.

Reviewer 2 Report

The authors present an application of an amphoteric terpolymer and a comparison with hydrolyzed polyacrylamide

I will start with an overview of the paper:

The paper is well written, although there is a serious concern regarding:

1.       The originality of the paper

2.       The discussion

3.       Fig. 12 of the present paper, and figure 9 of:

Gussenov, I.; Mukhametgazy, N.; Shakhvorostov, A.; Kudaibergenov, S. Synthesis and Characterization of High Molecular Weight Amphoteric Terpolymer Based on Acrylamide, 2-Acrylamido-2-Methyl-1-Propanesulfonic Acid Sodium Salt and (3-Acrylamidopropyl) trimethylammonium Chloride for Oil Recovery. Chemical Bulletin of Kazakh National University 2021, 4, 12-20. doi:10.15328/cb1243.

1.       The above paper shows a study by using the terpolymer used in this present paper, the novelty would be that there is a comparison with HPAM at 200 g/L

2.       The discussion in the present paper is quite challenging to understand, in the previous research there was a 0.2 g mass of produced oil at a salinity of 163 g/L doi:10.15328/cb1243 with the terpolymer. In the present paper, it is 0.7 g with HPAM (better performance than water which is reasonable). Nevertheless, the rationale of all the papers is that it is due to an HPAM decrease in viscosity at 200 g/L and no decrease in the terpolymer solution viscosity at this condition. In paper doi:10.15328/cb1243 there is a viscosity of 10.4 mPa/s at 163 g/L for HPAM and in the present paper a viscosity of 10.4 mPa/s at 200 g/L for the same system. There is a decrease of around 240 g/L in the present paper (and no experiments at this condition were performed, this is an excessive salinity where there could be salt-induced precipitation, formation of solid precipitates, etc)

3.       Figure 9 of doi:10.15328/cb1243 and 12 of the present paper show the ORF% at 163 g/L and 200 g/L of salt for the terpolymer 80-10-10, both ORF% are around 25% for both. Thus, how is it that for the present paper is not a good recovery, although for the previous one was a suitable one (although lower than HPAM)

Therefore, if it is not a viscosity-related phenomenon for HPAM between 163 and 200 g/L (by point 2. above), I don’t understand what the originality of the paper would be, the recovery at 200 g/L is low although higher than HPAM (2 times), although all the reasoning of the viscosity loss is flawed, due to 2.

I don’t see a real application regarding the cost to the HPAM comparison (2 times). How many times this polymer is more costly than HPAM? My main concern is that the discussion related to viscosity of HPAM is flawed.

Now I will perform detailed recommendations:

Is there an extended literature that call this polymers ATP??? If not, change the abbreviation. A single search shows that ATP is related to Adenosine triphosphate https://scholar.google.fr/scholar?hl=fr&as_sdt=0%2C5&q=atp+polymer+eor&btnG=

Introduction: the authors should extend further the previous work regarding this type of polymer mixtures and the mechanisms by which they work at high salinity. Include previous works with similar polymers.

Fig 1. The authors are working in the 15 to 20% range, this figure would be not suitable to show the concentration range you are working with. I would recommend modifying it to wt% and extending it to 20 % (you performed measurements at 20% for HPAM)

Line 90-91: “The present article compares the efficiency of the two polymers as regards their 90 rheological behavior and sand pack flooding.” The gap is outlined in the previous paragraphs86-90. I would recommend the authors to extend the objective and goal further.

Because of this, i.e., flawed discussion (it is not HPAM viscosity difference between 163 and 200 g/L salinity) and the inapplicability of the polymer in real reservoirs (due to the high cost, low recovery). I consider the research is not suitable for publication in polymers IF > 5

Best regards

Author Response

Reviewer 2

The authors present an application of an amphoteric terpolymer and a comparison with hydrolyzed polyacrylamide

I will start with an overview of the paper:

The paper is well written, although there is a serious concern regarding:

  1. The originality of the paper
  2. The discussion
  3. Fig. 12 of the present paper, and figure 9 of:

Gussenov, I.; Mukhametgazy, N.; Shakhvorostov, A.; Kudaibergenov, S. Synthesis and Characterization of High Molecular Weight Amphoteric Terpolymer Based on Acrylamide, 2-Acrylamido-2-Methyl-1-Propanesulfonic Acid Sodium Salt and (3-Acrylamidopropyl) trimethylammonium Chloride for Oil Recovery. Chemical Bulletin of Kazakh National University 2021, 4, 12-20. doi:10.15328/cb1243.

  1. The above paper shows a study by using the terpolymer used in this present paper, the novelty would be that there is a comparison with HPAM at 200 g/L
  2. The discussion in the present paper is quite challenging to understand, in the previous research there was a 0.2 g mass of produced oil at a salinity of 163 g/L doi:10.15328/cb1243 with the terpolymer. In the present paper, it is 0.7 g with HPAM (better performance than water which is reasonable). Nevertheless, the rationale of all the papers is that it is due to an HPAM decrease in viscosity at 200 g/L and no decrease in the terpolymer solution viscosity at this condition. (1) In paper doi:10.15328/cb1243 there is a viscosity of 10.4 mPa/s at 163 g/L for HPAM and in the present paper a viscosity of 10.4 mPa/s at 200 g/L for the same system. (2) There is a decrease of around 240 g/L in the present paper (and no experiments at this condition were performed, this is an excessive salinity where there could be salt-induced precipitation, formation of solid precipitates, etc)

Answer:

(1) The paper [22] – doi:10.15328/cb1243, which was mentioned by the reviewer does not provide any results on HPAM viscosity. The viscosity of 10.4mPa*sec is mentioned only in [21] DOI:10.31489/2020Ch4/119-127, which is the viscosity value of the terpolymer, not that of HPAM.

(2) Yes, figure 4 in the present paper shows that the viscosity of HPAM reduces very fast when the salinity increases from 200 to 240 g/L. This fact indicates that 200 g/L might be a critical point in salinity beyond which the application of HPAM is not recommended and our findings demonstrate this quantitatively by the results of viscosity measurements and sand pack flooding tests.

Figure 4. Dynamic viscosities of 0.25 wt.% TPA (curves 1,3) and HPAM (curves 2,4) in a saline solution at 24 °C (curves 1,2) and 60 °C (curves 3,4), with a shear rate of 7.32 s-1

  1. Figure 9 of doi:10.15328/cb1243 and 12 of the present paper show the ORF% at 163 g/L and 200 g/L of salt for the terpolymer 80-10-10, both ORF% are around 25% for both. Thus, how is it that for the present paper is not a good recovery, although for the previous one was a suitable one (although lower than HPAM)

Therefore, if it is not a viscosity-related phenomenon for HPAM between 163 and 200 g/L (by point 2. above), I don’t understand what the originality of the paper would be, the recovery at 200 g/L is low although higher than HPAM (2 times), although all the reasoning of the viscosity loss is flawed, due to 2.

I don’t see a real application regarding the cost to the HPAM comparison (2 times). How many times this polymer is more costly than HPAM? My main concern is that the discussion related to viscosity of HPAM is flawed.

Answer:

Fig. 9 of the paper [22] doi:10.15328/cb1243 shows the dynamic viscosity, but not ORF. Probably the reviewer meant fig.9 of paper [21]

[21] DOI:10.31489/2020Ch4/119-127                     Current paper

Please, note that Fig.9 [21] presents the total ORF obtained by the injection of water and polymer, whereas Fig.12 of the current paper presents the ORF obtained only by the polymer injection after water flooding. We do not compare fig.9 with fig.12 simply because in these experiments different polymer concentrations, oil viscosities, brine salinities and porous media models were used.

With regards to the cost of the polymer, unfortunately, we did not evaluate the cost of ternary polyampholyte AAm-AMPS-APTAC in comparison with the HPAM because it is out of scope of our subject.

With regards to the viscosity of HPAM, polymer flooding provides additional EOR due to the higher viscosity of the polymer solution in comparison with water. The increase of viscosity provides a lower mobility factor in the porous media which reduces water fingering and ensures a more stable oil displacement front. As seen in figure 4, beyond 200 g/L the viscosity of HPAM starts to reduce sharply. This indicates that 200 g/L is a critical for HPAM in the vicinity of which the polymer may fail to provide high oil recovery in comparison with other more salt-tolerant polymers. Likely, our sand pack flooding results support this assumption.

Now I will perform detailed recommendations:

Is there an extended literature that call this polymers ATP??? If not, change the abbreviation. A single search shows that ATP is related to Adenosine triphosphate https://scholar.google.fr/scholar?hl=fr&as_sdt=0%2C5&q=TPA+polymer+eor&btnG=

Answer: The abbreviation of amphoteric terpolymer (ATP) was replaced by ternary polyampholyte (TPA) throughout the text.

Introduction: the authors should extend further the previous work regarding this type of polymer mixtures and the mechanisms by which they work at high salinity. Include previous works with similar polymers.

Answer: According to reviewer comments we considerably expanded the Introduction and added appropriate literature refs to explain the mechanism of action of our polyampholyte at high salinity. The updated text is highlighted in red.

Fig 1. The authors are working in the 15 to 20% range, this figure would be not suitable to show the concentration range you are working with. I would recommend modifying it to wt.% and extending it to 20 % (you performed measurements at 20% for HPAM)

Answer:

Figure 1 was adapted from the literature. This figure is included into the introduction section just to demonstrate that the viscosity of HPAM is very sensitive to salinity increases. Fig.1 has been corrected.

Line 90-91: “The present article compares the efficiency of the two polymers as regards their 90 rheological behavior and sand pack flooding.” The gap is outlined in the previous paragraphs 86-90. I would recommend the authors to extend the objective and goal further.

Answer:

We added two paragraphs to extend the objective and goal of our research.

Because of this, i.e., flawed discussion (it is not HPAM viscosity difference between 163 and 200 g/L salinity) and the inapplicability of the polymer in real reservoirs (due to the high cost, low recovery). I consider the research is not suitable for publication in polymers IF > 5

Answer:

The research results clearly show that HPAM fails to provide high viscosity values at salinities beyond 200 g/L. On the contrary the viscosity of the terpolymer is not negatively affected by the increase of salinity beyond 200 g/L. The sand pack flooding results demonstrate that HPAM is not a reliable choice at 200 g/L. This is a relatively novel finding which will attract the attention of enhanced oil recovery engineers and scientists.

For example, fig. 8 of the current paper shows that HPAM is better than amphoteric terpolymer when the salinity was 163 g/L.

However, when the salinity was 200 g/L the amphoteric terpolymer outperformed the HPAM (fig.10 below).

We relate this difference between HPAM and TPA performances to the salinity of brine. Beyond 200 g/L, the viscosity of HPAM reduces very fast (our measurements show this), which is why oil recovery performance of HPAM at and beyond 200 g/L is questionable.

Reviewer 3 Report

The authors developed a comparative study of oil recovery using amphoteric terpolymer and hydrolyzed polyacrylamide. 

A very important aspect of the manuscript must be the novelty of the work. Give a strong reason why the work is novel.

I found one work very close to yours and there is no mention of it in your manuscript. State the reason why it is different, or include it in the manuscript as a reference. https://onlinelibrary.wiley.com/doi/10.1002/cjce.24300

Make sure that the language of the manuscript is improved and is more readable.

The abstract is very long and complicated. Simplify and write a clear abstract.

Introduction:

Heavy oil and its improvement possibilities should be mentioned in introduction. Other enhancement methods should also be compared in the introduction. One of them is aquathermolysis. This would clearly improve the introduction and further emphasize the advantages of your studied method. https://www.sciencedirect.com/science/article/pii/S0016236121017506

Line 63-75: Another important work related to your topic via polyacrylamide. https://www.mdpi.com/2073-4360/12/3/708

Line 76-91: it is written very complicated. Improve it.

At the end of the introduction, emphasize the importance of your work as well as the perspective.

Experimental part:

Line 99: State the conditions under which this synthesis or reaction can reliably take place.

Line 109: The information from Table 1 could also be included in the main text, as it contains very little information.

Describe the materials used and especially the methods in more detail. This means the instrument settings, the way the analysis was performed, and so on.

Line 157: Figure 4 needs to be improved to make it more attractive.

Line 172: It is necessary to add measurement deviations to table 3.

Figure 9: So many points next to each other are very poorly tracked.

Figure 11 is extremely confusing. Please find a way to improve it.

Author Response

Reviewer 3

The authors developed a comparative study of oil recovery using amphoteric terpolymer and hydrolyzed polyacrylamide.

A very important aspect of the manuscript must be the novelty of the work. Give a strong reason why the work is novel.

Answer:

The novelty of our research is at the end of the Introduction. Please see text highlighted in red.

I found one work very close to yours and there is no mention of it in your manuscript. State the reason why it is different, or include it in the manuscript as a reference. https://onlinelibrary.wiley.com/doi/10.1002/cjce.24300

Answer:

The article has been included in the reference list, and mentioned in the text. Please see page 12, lines 315-317.

Make sure that the language of the manuscript is improved and is more readable.

The abstract is very long and complicated. Simplify and write a clear abstract.

Answer:

The abstract has been corrected.

Introduction:

Heavy oil and its improvement possibilities should be mentioned in introduction. Other enhancement methods should also be compared in the introduction. One of them is aquathermolysis. This would clearly improve the introduction and further emphasize the advantages of your studied method. https://www.sciencedirect.com/science/article/pii/S0016236121017506

Line 63-75: Another important work related to your topic via polyacrylamide. https://www.mdpi.com/2073-4360/12/3/708

Answer:

These papers have been mentioned in the introduction section. Please see page 4, line 127-132.

Line 76-91: it is written very complicated. Improve it.

At the end of the introduction, emphasize the importance of your work as well as the perspective.

Answer:

The text has been corrected. Please see the parts highlighted in red.

Experimental part:

Line 99: State the conditions under which this synthesis or reaction can reliably take place.

Answer:

The synthesis has been briefly described. Please see page 4-5, lines 142-150.

Line 109: The information from Table 1 could also be included in the main text, as it contains very little information.

Describe the materials used and especially the methods in more detail. This means the instrument settings, the way the analysis was performed, and so on.

Answer:

Table 1 has been included in the main text.

The additional information was added, please see the text highlighted in red in the Experimental section.

Line 157: Figure 4 needs to be improved to make it more attractive.

Answer

Figure 4 has become figure 5 and was improved. In particular, we have changed the colors of the markers so that they match with the colors of the lines. Moreover, each line has a differently shaped marker, so that if the paper is printed in black and white, a reader will still be able to distinguish between the lines.

Line 172: It is necessary to add measurement deviations to table 3.

Answer:

Has been corrected. However, Table 3 has become Table 2.

Figure 9: So many points next to each other are very poorly tracked.

Answer:

Figure 9 has become figure 10, and has been corrected.

Figure 11 is extremely confusing. Please find a way to improve it.

Answer:

Figure 11 has become figure 12, and has been corrected.

Round 2

Reviewer 2 Report

The authors performed the recommended revisions and responded to all my questions.

Best regards

Reviewer 3 Report

The manuscript has been significantly improved. My questions were answered. I think the manuscript can be accepted in its current form.